# 3D-Prover: Diversity Driven Theorem Proving With Determinantal Point Processes

**Sean Lamont[1,2], Christian Walder[3], Amir Dezfouli[4], Paul Montague[2], Michael Norrish[1]**
[1]Australian National University
[2]Defence Science and Technology Group
[3]Google DeepMind
[4]BIMLOGIQ
`sean.lamont@anu.edu.au`

## Abstract

A key challenge in automated formal reasoning is the intractable search space, which grows exponentially with the depth of the proof. This branching is caused by the large number of candidate proof tactics which can be applied to a given goal. Nonetheless, many of these tactics are semantically similar or lead to an execution error, wasting valuable resources in both cases. We address the problem of effectively pruning this search, using only synthetic data generated from previous proof attempts. We first demonstrate that it is possible to generate semantically aware tactic representations which capture the effect on the proving environment, likelihood of success, and execution time. We then propose a novel filtering mechanism which leverages these representations to select semantically diverse and high quality tactics, using Determinantal Point Processes. Our approach, 3D-Prover, is designed to be general, and to augment any underlying tactic generator. We demonstrate the effectiveness of 3D-Prover on the miniF2F and LeanDojo benchmarks by augmenting popular open source proving LLMs. We show that our approach leads to an increase in the overall proof rate, as well as a significant improvement in the tactic success rate, execution time and diversity. We make our code available at `https://github.com/sean-lamont/3D-Prover`.

## 1 Introduction

Interactive Theorem Proving (ITP) traditionally involves a human guiding an ITP system to verify a formal proposition. The applications range from secure software (Tan et al., 2019) to the verification of mathematical results (Hales et al., 2017). There has been significant interest in automating this process, with formalization efforts requiring a high level of human expertise (Klein et al., 2009). It is also considered a 'grand challenge' for AI, requiring a high level of reasoning and planning to be successful (Reddy, 1988). Even large general purpose models struggle with the complexity of the task, with for example GPT-4 only able to solve 13.5% (Thakur et al., 2023) of the high school level miniF2F-test (Zheng et al., 2021) benchmark. This has motivated specialized models and search algorithms to address the unique challenges of the domain (see e.g. (Li et al., 2024) for a review).

With most non-trivial proofs requiring long chains of correct reasoning, it is a challenge to generate them in one pass without mistakes. The addition of a search algorithm is common for addressing this (Xin et al., 2024; Wu et al., 2024). Under this paradigm, candidate tactics are generated and executed in the proving system, which (if successful) results in new subgoals to prove. This generates a tree of possible proof paths, where a search algorithm selects the most promising nodes to expand. The primary challenge faced by these approaches is the exponential growth in the number of proof paths, limiting the complexity of the problems that can be solved efficiently. Many of the generated

39th Conference on Neural Information Processing Systems (NeurIPS 2025).

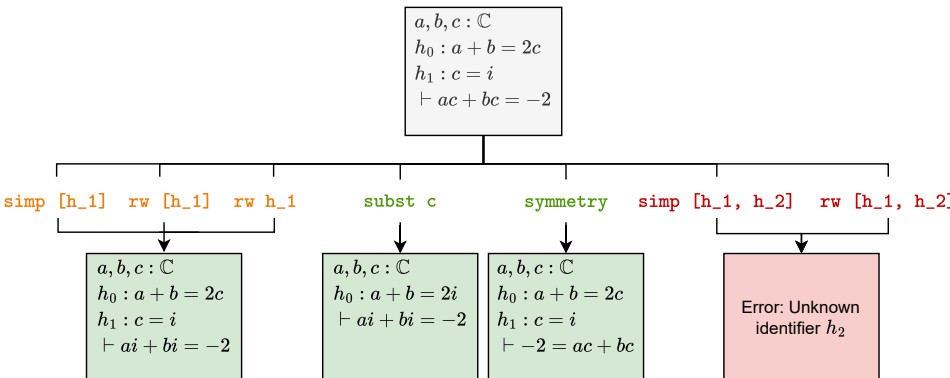

Figure 1: An example node expansion for a failed ReProver attempt, which 3D-Prover was able to prove. Tactics on the left result in the same proof state, tactics on the right result in an error, and tactics in the centre result in a unique proof state. The high error rate and tactic similarity motivates our filtering approach, which prunes the search space to give a diverse set of subgoals.

tactics are equivalent, modulo variable renaming and other semantics-preserving transformations. See Figure 1 for a sample search tree from ReProver (Yang et al., 2023), where several semantically similar paths are explored, wasting valuable resources. Simple lexical similarity scores fail to cover the semantics (meaning) of a tactic, as captured by the effect of the tactic on the environment. For example, an expression and its negation vary by only a single character, but have a large semantic difference. It is therefore desirable to filter tactics by their semantic rather than syntactic diversity. In addition, many tactics lead to an execution error from the prover. From our experiments with miniF2F, we find approximately 75-85% of tactics result in an execution error (Section 2.2). As tactic execution can be expensive, this further restricts the space of proofs which can be explored efficiently.

These challenges motivate our proposed approach, **D**iversity **D**riven **D**eterminantal **P**oint **P**rocess **P**rover (3D-Prover). 3D-Prover adds an extra 'dimension' to existing proving systems by including a filtering mechanism on top of the existing tactic generation and search components. 3D-Prover uses Determinantal Point Processes (Kulesza, 2012) to prune the search space by filtering tactics to diverse and high quality subsets. The rich synthetic data generated from proof attempts enables us to learn the effect tactics have on the environment, including the error likelihood and execution time. We leverage this to generate tactic representations which reflect their semantics, which 3D-Prover uses to filter tactics based on a combination of their diversity and quality. 3D-Prover allows for a direct tradeoff between search objectives, with hyperparameters controlling the weighting of error, time and diversity in the filtering process. 3D-Prover is a general approach which can be used to augment any underlying tactic generator. We demonstrate this by augmenting the popular ReProver and InternLM2.5-Step-Prover LLMs to obtain a significant improvement in the success rate, execution time and diversity of tactics, and the overall proof success rate. To summarise our contributions:

- We study the feasibility of learning the environment dynamics of proving systems. We demonstrate tactic representations which capture the likely effect on the environment, using them to predict the likelihood of success and execution time of a tactic, as well as the resulting proof state or error message.
- We propose a novel edge filtering approach using Determinantal Point Processes (Kulesza & Taskar, 2011), which leverage these representations to select semantically diverse subsets of quality tactics. Our method is modular and can be used with any tactic generator.
- We evaluate our approach by augmenting ReProver (Yang et al., 2023) on the miniF2F (Zheng et al., 2021) benchmarks, where we demonstrate a significant improvement in the tactic success rate, diversity and overall proof rate.

## 1.1 Related work

There is little prior work on learning the effect of a tactic on the proving environment, with only Xin et al. (2024) using successful environment responses as an auxiliary objective. We investigate the task in detail, as well as learning the error likelihood, error messages and execution time, which

we use to generate useful tactic representations. Several approaches use previous proof attempts to improve performance, using the sparse binary signal from the proof result (Li et al., 2024). This has been used to improve search (Lample et al., 2022; Wang et al., 2023), however these approaches do not consider node diversity, with nothing preventing the exploration of semantically similar paths. First & Brun (2022) examine a diverse ensemble of tactic models, whereas we focus on diversity with respect to the search, given an arbitrary underlying tactic model (or models). Recently, Yang et al. (2025) select subgoals based on their diversity, with a simple embedding model over the subgoal text. Our approach does not need to execute the tactics, as we learn embeddings reflecting the environment dynamics and use these to select tactics before execution, thereby saving resources.

## 1.2 Background: Determinantal Point Processes

Determinantal Point Processes (DPPs) are a class of probabilistic models for sampling subsets from a ground set $\mathcal{Y}$. They provide an inherent trade-off between the diversity and quality of the sampled subsets, successfully being applied to this end across a variety of domains (Kulesza, 2012; Hsiao & Grauman, 2018; Zhang et al., 2016). This motivates their use in our filtering approach (Section 3).

In line with Kulesza (2012), for $|\mathcal{Y}| = n$ we define the kernel $L \in \mathbb{R}^{n \times n}$ of a DPP as the Gram matrix $L = B^T B$ for $B \in \mathbb{R}^{n \times d}$, where row $\boldsymbol{b}_i \in \mathbb{R}^d$ of $B$ represents element $i \in \{1, \ldots, n\}$ of $\mathcal{Y}$. The $\boldsymbol{b}_i$ are commonly decomposed into a set of unit norm diversity features $\boldsymbol{\phi}_i \in \mathbb{R}^d$ and quality scores $q_i \in \mathbb{R}^+$, so that $\boldsymbol{b}_i = q_i \boldsymbol{\phi}_i$, $||\boldsymbol{\phi}_i|| = 1$ for all $i \in \{1, \ldots, n\}$. The similarity matrix $S$ is then defined as $S_{ij} = \boldsymbol{\phi}_i^T \boldsymbol{\phi}_j$. The probability of sampling $A \subseteq \mathcal{Y}$ is then proportional to the determinant of the submatrix of $L$ indexed by $A$, $\mathbb{P}(A) \propto \det(L_A) = (\prod_{i \in A} q_i^2)\det(S_A)$. Geometrically, this determinant is the volume of the parallelepiped spanned by the submatrix $L_A$, which as we see in Figure 3, is maximised based on a combination of the similarity and length (quality) of the chosen elements. In this way, DPPs elegantly trade off between the quality and diversity of elements. Normally the size of the sampled subset $|A|$ is variable, however Kulesza & Taskar (2011) introduce $k$-DPPs which restricts the size of the subset to a fixed $k \in \mathbb{N}$, and where the probability of sampling $A$ is normalised over subsets of size $k$. That is, for a $k$-DPP, $\mathbb{P}(A) = \det(L_A)/\sum_{|A'|=k} \det(L_{A'})$.

## 2 Transition Aware Representation Learning

One proof attempt can generate large amounts of data. We found a single pass of ReProver on the miniF2F-valid benchmark of 244 proofs results in approximately 500,000 transitions, capturing rich information about the error likelihood, execution time and resulting proof state or error message. We now explore the feasibility of using this data to learn how tactics affect the environment, operationalising this as a supervised learning task: given a goal and tactic, we predict the error status, execution time and environment output. We effectively learn these targets from only this synthetic data, and embed this information into a compact tactic representation. The upshot, as we show in Section 3, is that this can be used to improve the performance of subsequent proof attempts.

### 2.1 Transition Models

The result of a proof attempt (formalised in A) is the dataset $\mathcal{D}$ of *transitions* $\{(g, t, s, \tau, o)\}$, which captures the results of applying *tactics* $t \in \mathcal{T}$ to *goals* $g \in \mathcal{S}$. The *status* $s \in \{0, 1\}$, indicates a success (1) or failure (0), $\tau \in \mathbb{R}$ gives the execution time and the *output* $o \in \mathcal{O}$ is the environment response, which is an error message, new subgoals, or a proof success. We propose a method to learn tactic representations $\boldsymbol{e} \in \mathbb{R}^d$ which capture the result $(s, \tau, o)$ of applying $t$ to $g$. By using these as features for DPP, we can filter tactics based on their expected outcome, before they are executed.

We define our *transition model* $\xi : \mathcal{S} \times \mathcal{T} \to [0, 1] \times \mathbb{R} \times \mathcal{O}$ as a mapping from a goal $g$ and tactic $t$ to an estimate of the status $s$, time $\tau$ and output $o$. To ensure $\xi$ admits effective representations in $\boldsymbol{e}$, we construct it as follows. The Encoder $E : \mathcal{S} \times \mathcal{T} \to \mathbb{R}^d$ takes the goal $g$ and tactic $t$ as input, and outputs our representation $E(g, t) = \boldsymbol{e}$. As $\boldsymbol{e}$ will be used as the diversity feature for DPP, it is constrained to unit norm $||\boldsymbol{e}|| = 1$. The Predictor $P : \mathbb{R}^d \to [0, 1] \times \mathbb{R}$ maps $\boldsymbol{e}$ to an error probability for the status and a score for the time prediction, with $P(\boldsymbol{e}) = (\hat{s}, \hat{\tau})$. The Decoder $D : \mathbb{R}^d \times \mathcal{S} \to \mathcal{O}$ maps $\boldsymbol{e}$ and $g$ to the output prediction, such that $D(\boldsymbol{e}, g) = \hat{o}$. The transition model is then

$$\xi(g, t) = (P(E(g, t)), D(E(g, t), g)) = (\hat{s}, \hat{\tau}, \hat{o}). \tag{1}$$

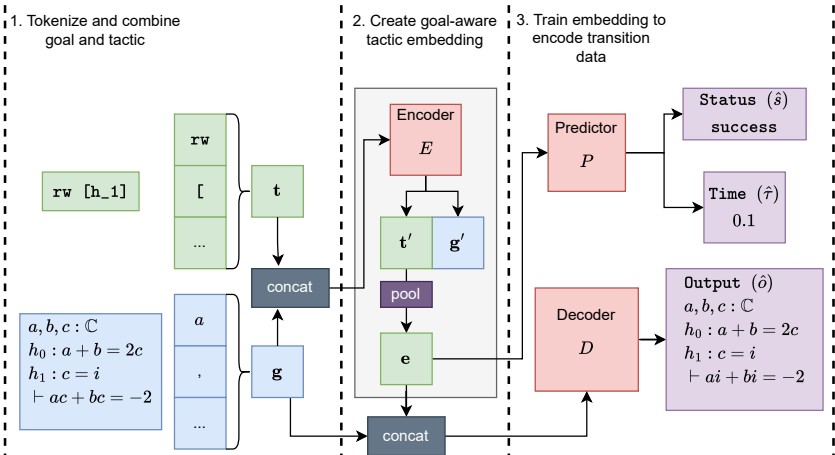

Figure 2: Our COMBINED architecture for learning transition aware tactic embeddings. The tactic $\mathbf{t}$ and goal $\mathbf{g}$ are concatenated and passed through the Encoder $E$. A representation vector $\mathbf{e}$ is generated by mean-pooling over the tactic token embeddings $\mathbf{t}'$. The Predictor $P$ takes this embedding and predicts whether the tactic results in an error (Status), and the execution time (Time). The Decoder $D$ takes the embedding and goal to predict the environment response (Output), which is either an error message or new goals to prove. This setup yields a compact representation of the tactic which captures its effect on the proving environment, enabling our proposed filtering model.

We note that the Decoder and Predictor can only access information of $t$ through $e$. Hence our architecture requires the Encoder to learn an effective representation for $e$, so that the Decoder and Predictor can use this to determine the subsequent effect of the tactic on the environment.

## 2.2 Experiments

For our experiments, we use an Encoder-Decoder Transformer for the Decoder $D$, and an Encoder-Only Transformer for the Encoder $E$. We take the pretrained ReProver (Yang et al., 2023) LLM to initialise both components. We implement the Predictor $P$ as a single hidden layer MLP, with hidden dimension $d/2$ (where $d = 1472$) and two real valued output nodes. The time prediction $\hat{\tau}$ is the output of the first node, and the status prediction $\hat{s}$ is taken as the sigmoid of the second. We use this simple Predictor architecture to speed up our filtering algorithm presented in Section 3.

We investigate several variations of the transition model $\xi$. For the **COMBINED** model (Figure 2), the tactic is concatenated with the goal, and the embeddings from the Encoder are computed for all tokens. We then generate a single tactic embedding by mean-pooling over the tactic tokens. We examine the COMBINED model both with the full goal text, and a variation **COMBINED (SMALL GOAL)** which embeds the goal first, and concatenates it as a single token vector to the tactic. This variation allows for more efficient batching when used for filtering in Section 1, as the goal need only be embedded once for multiple tactics, however gives less information to the model. The **SEPARATE** model encodes the tactic without attending to the goal. We hypothesise that allowing the tactic tokens to attend to the goal will allow the Encoder to better represent the semantics of the tactic. To form a naive baseline, we implement a **NO TACTIC** model which does not use the tactic at all, and instead uses only the goal tokens. We do this to account for any inherent patterns in the goal which may be predictive of the outcome, for example a particular goal which has a high error rate. This allows us to ground our results in the performance of this baseline, so we can observe the direct effect of the tactic in predictive performance. We also compare with an **ALL TOKENS** model which uses all tactic tokens for the Decoder without reducing to a single embedding. We implement this comparison to see the degree of information loss induced by reducing tactics to a single vector. Given $\alpha_s, \alpha_\tau, \alpha_o \in \mathbb{R}^+$, with estimates $\hat{s}, \hat{\tau}, \hat{o}$ and for minibatch $\mathcal{B} \subseteq \mathcal{D}$, we optimise the following:

$$\sum_{(g,t,s,\tau,o)\in\mathcal{B}} \alpha_s \mathcal{L}_s(s, \hat{s}) + \alpha_\tau \mathcal{L}_\tau(\tau, \hat{\tau}) + \alpha_o \mathcal{L}_o(o, \hat{o}). \tag{2}$$

Table 1: Results for predicting unseen environment responses given a goal and tactic, for transitions from miniF2F-valid. The No Tactic result forms a baseline to assess the impact of the tactic representation. We observe that any tactic representation enables far better predictions, and constraining these to a single vector (Combined and Separate) does not hurt the performance gain. This demonstrates tactic representations which capture their effect on the environment, enabling our filtering model in Section 3. Comparing the Combined and Separate models, allowing the representation to attend to the goal leads to a large improvement.

| | Output | | | Status | | | Time |
|---|---|---|---|---|---|---|---|
| Embedding | BLEU | ROUGE-L F1 | Top-4 | F1 | TPR | TNR | MSE |
| All Tokens | 0.31 | 0.38 | 0.31 | 0.85 | 0.82 | 0.96 | 0.17 |
| Combined | **0.33** | **0.39** | **0.32** | **0.88** | **0.85** | **0.97** | **0.16** |
| Combined (Small Goal) | 0.30 | 0.36 | 0.29 | 0.85 | 0.81 | 0.96 | 0.20 |
| Separate | 0.27 | 0.34 | 0.27 | 0.76 | 0.71 | 0.94 | 0.28 |
| No Tactic | 0.17 | 0.22 | 0.13 | 0.22 | 0.14 | 0.96 | 0.37 |

The hyperparameters $\alpha_s, \alpha_\tau, \alpha_o$ control the weighting of the status, time and output losses. For simplicity, we set these to 1, however they could be tuned to reflect the relative importance of each task. We use the binary cross-entropy loss $\mathcal{L}_s$ for the status prediction, the mean squared error (MSE) $\mathcal{L}_\tau$ for the time prediction, and the cross-entropy loss $\mathcal{L}_o$ for the output prediction.

We obtain $\mathcal{D}$ from a single ReProver attempt on miniF2F-valid, yielding 498,236 transitions split randomly into 95% training, 5% testing. For the error prediction task, we reweight classes to account for imbalance, which is approximately 75% error, 25% success. We use the AdamW optimizer, with a learning rate of $10^{-5}$ and a batch size of 1, train for 2 epochs, and report the results on the test set. To assess the Output prediction, we generate 4 outputs with beam search for each transition. We use BLEU (Papineni et al., 2002) and ROUGE-L (Lin, 2004) to assess the quality of the highest scoring beam in comparison to the ground truth, which is either an error message or new subgoals. The Top-4 accuracy is the proportion of samples with one beam identical to the ground truth. For Status, we take the prediction as 1 if $\hat{s}_{ki} > 0.5$ and 0 otherwise, reporting the F1 score, true positive rate (TPR) and true negative rate (TNR). For time, we take the Mean Squared Error (MSE) of the prediction.

Table 1 summarises the performance of our transition models. Our results suggest tactic representations which capture useful information about their effect on the environment [1], which we can see by the clear improvement across all approaches compared to the No Tactic baseline. The improvement of Combined over Separate supports our hypothesis that we can better predict transitions when the tactic embedding attends to the goal. As expected, Combined outperforms Combined (Small Goal), with Combined (Small Goal) significantly outperforming the Separate model. Combined (Small Goal) therefore gives an effective compromise between the accurate but expensive Combined model, and the goal-unaware Separate model. We note the All Tokens model, with the Decoder attending to the full tactic, does not improve upon the full Combined model. This shows our architecture effectively represents the tactic as a single embedding without losing any relevant information. These results are the first to demonstrate the feasibility of learning the environment dynamics of proof systems. To illustrate the difficulty of this task, all predictions for the Combined model and their ground truth are provided with our code.

## 3 3D-Prover

Algorithm 1 defines 3D-Prover, which maps tactics $T$ from the underlying tactic policy $\pi_0$ to a subset $T'$ of size $K$. We use the Encoder $E$ and Predictor $P$ from Section 2 to generate tactic embeddings $\phi_i$ and predict the time and error likelihood. As they are unit norm, the embeddings $\phi_i$ encode the predicted environment response through their direction. The quality score $q_i$ then scales $\phi_i$ based on the tactic model logits $m_i$, as well as the predicted error likelihood $s_i$ and execution time $\tau_i$. We have hyperparameters for normalisation temperature $\theta$, as well as error and time weights

---

[1]See Appendix E for further analysis of these embeddings.

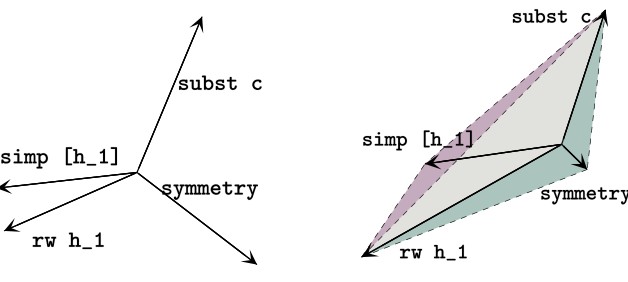

(a) Initial tactic embeddings $\phi_i$, representing predicted outcome.

(b) Quality scaled embeddings, $q_i\phi_i$, to be filtered by $k$-DPP

Figure 3: Visualisation of DPP for tactic filtering. The tactic embeddings from the transition model are scaled by quality scores, before a subset of tactics are selected using $k$-DPP. Subsets are chosen proportionally to the area spanned by their elements, giving a combination of quality and diversity. For this simplified example, we take the 2D PCA projection of embeddings for tactics in Figure 1, setting the quality to the scaled generator logits. Comparing the shaded areas in (b) and assuming `subst c` and `rw h₁` have been selected, we see that `symmetry` is favoured over `simp [h₁]`. Although `simp [h₁]` is scored higher by the generator, it is less diverse with respect to `subst c` and `rw h₁`.

---

**Algorithm 1:** 3D-Prover

**Input** : Goal $g$, candidate tactics $T = \{t_i\}_{i=1}^N$, filter size $K$, Encoder $E$, Predictor $P$, error weight $\lambda_s$, time weight $\lambda_\tau$, temperature $\theta$, tactic policy $\pi_0$

**Output :** Filtered tactics $T' \subset T$

```
// Compute embeddings and scores for each tactic
```
**for** $i$ *in* $\{1, \ldots, N\}$ **do**

    $\phi_i \leftarrow E(g, t_i)$

    $(s_i, \tau_i) \leftarrow P(\phi_i)$

    $\tau_i \leftarrow 1 - \frac{\tau_i}{||\boldsymbol{\tau}||}, \boldsymbol{\tau} = (\tau_1, .., \tau_N)$

    $m_i \leftarrow \frac{\exp(\pi_0(t_i|g)/\theta)}{\sum_{j=1}^N \exp(\pi_0(t_j|g)/\theta)}$

    $q_i \leftarrow m_i + \lambda_s s_i + \lambda_\tau \tau_i$

```
// Compute kernel matrix
```
$L \leftarrow B^T B$, where $B = [q_1\phi_1, \ldots, q_N\phi_N]$

Compute eigenvalues $\lambda_i$ and eigenvectors $\boldsymbol{v}_i$ of $L$

Sample $J \subset \{1, \ldots, N\}$ using Algorithm 2 of Kulesza & Taskar (2011), with $\{(\boldsymbol{v}_i, \lambda_i)\}$, $k = K$

**return** $T' = \{t_j\}_{j \in J}$

---

$(\lambda_s, \lambda_\tau)$. $\theta$ controls the scaling temperature of the model logits, with a higher temperature flattening the distribution. It therefore adjusts the diversity bias of 3D-Prover by reducing the impact of the quality scores when sampling. We then compute the kernel $L$ from $q_i$ and $\phi_i$, and sample a subset of tactics $T'$ using the $k$-DPP algorithm (Kulesza & Taskar, 2011). Figure 3 visualises this process, where tactics subsets are sampled in proportion to their shaded area.

### 3.1 Experiments

We test the performance of 3D-Prover with two setups. We first use ReProver (Yang et al., 2023) as the underlying tactic policy $\pi_0$, as it is a small ($\sim 300$M parameters) and popular open source proving model, allowing us to run extensive experiments and ablations in a reasonable time frame. To evaluate our approach over a large (7B) state-of-the-art model, we also present a smaller scale experiment using InternLM2.5-Step-Prover (Wu et al., 2024). We run our experiments in Lean (De Moura et al., 2015) using the BAIT (Lamont et al., 2024) platform with a modified LeanDojo (Yang et al., 2023) environment, where we set an environment timeout of 600 seconds per proof attempt. We train a COMBINED model for Reprover and a COMBINED (SMALL GOAL) model for InternLM

Table 2: Pass@1 results on miniF2F, with $K$ tactics selected per node from ReProver. 3D-Prover uses a transition model trained from miniF2F-valid transitions. For miniF2F-test, we report the mean along with minumum and maximum over four runs, noting that Top-$K$ is deterministic given ReProver uses Beam Search. The Gain column reports the relative improvement over the Top-$K$ baseline. Results for No Filtering were 27.8% for miniF2F-test and 27.9% for miniF2F-valid. We observe a clear improvement using 3D-Prover, which increases as more filtering is applied (lower $K$). Our results on miniF2F-test show that 3D-Prover can improve search even for proofs out of distribution of the transition model.

| $K$ | Top-$K$ | Random | 3D-Prover | Gain |
|---|---|---|---|---|
| *miniF2F-test (mean, minimum and maximum over four runs)* | | | | |
| 8 | 22.4 | 19.0 (18.4, 19.6) | **24.4 (23.7, 24.9)** | +8.9% |
| 16 | 26.5 | 25.4 (24.5, 25.7) | **27.3 (26.9, 27.8)** | +3.0% |
| 32 | 27.8 | 27.4 (26.9, 28.2) | **28.2 (27.3, 28.6)** | +1.4% |
| *miniF2F-valid (single run)* | | | | |
| 8 | 21.7 | 19.3 | **25.0** | +15.2% |
| 16 | 26.6 | 24.2 | **29.1** | +9.4% |
| 32 | 27.9 | 27.5 | **28.7** | +2.9% |

from Section 2.2, using transitions from their respective base models. The Encoder and Predictor components then generate tactic embeddings and quality scores as per Algorithm 1. We first examine the performance of 3D-Prover without hyperparameter tuning, setting $\lambda_s = \lambda_\tau = 0$, $\theta = 1$. We then perform ablation studies ( 3.1.2) with ReProver over miniF2F-valid to examine the influence of the hyperparameters on the tactic success rate, execution time and diversity of the environment response. For miniF2F-test, we allow the model multiple attempts per proof to increase confidence in the results, while for miniF2F-valid we allow one attempt per configuration to allow a wider set of ablations. We also present an additional experiment over the larger LeanDojo benchmark in Appendix D.

The default search policy is set to be Best First Search (BFS), with nodes expanded in order of their cumulative log probability. For each node selected, we generate $N = 64$ candidate tactics from the underlying model [2]. Following the original implementations, we use beam search for ReProver and sampling with $T = 0.7$ for InternLM. These form the ground set for the node, to be sub-sampled by the filtering algorithm. As beam search decoding is deterministic, the ground set for a given node is fixed across runs, allowing us to better isolate and compare approaches. We maintain sampling for InternLM to represent the original deployment scenario and test our approach under realistic usage. The filtering algorithm returns $K$ tactics, which are executed in the environment before updating the proof tree. For ReProver, we test three levels of filtering, with $K \in \{8, 16, 32\}$. Lower $K$ corresponds to more filtering, for which the filtering algorithm will have a greater impact. We compare 3D-Prover, as outlined in Algorithm 1, with three baselines. The **No Filtering** baseline represents the original approach with no filtering. The **Top-K** baseline takes the top $K$ tactics from the ground set as judged by their log probabilities, corresponding to the top $K$ beams. We take $K$ tactics at random from the ground set to form the **Random** baseline, as an exploration-focused comparison. For InternLM, we test $K = 8$ with the Top-$K$ and No Filtering baselines, and perform an additional experiment with Critic Guided search from (Wu et al., 2024) in place of BFS.

### 3.1.1 Proof Performance

Table 2 and 3 shows the results of our ReProver and InternLM experiments on miniF2F. We observe 3D-Prover outperforming every baseline over both models. The influence of filtering becomes more apparent as $K$ is decreased as there are more tactics filtered out. Reflecting this, the magnitude of improvement given by 3D-Prover increases for lower $K$. 3D-Prover is able to outperform both baselines by providing a tradeoff between the quality, as represented by Top-$K$, and the diversity of the tactics. The choice of $K$ also controls the depth of the proof search, with larger $K$ giving broader search, and smaller $K$ deeper search. As most discovered proofs are short (favouring broad search), we see the Pass@1 performance for lower values of $K$ is generally lower, however over multiple attempts it can be beneficial to use deeper searches (see Appendix B). Our InternLM results

---

[2]As InternLM is sampling, this is *up to* 64. See Appendix H for the distribution of tactic counts per node

Table 3: Results for miniF2F-test using tactics selected from InternLM2.5-Step-Prover (mean, maximum and minimum for Pass@1). We compare results using standard Best First Search model (BFS) and with the InternLM2.5 Critic Guided model for goal selection. 3D-Prover uses a transition model trained from transitions on miniF2F-test with the No Filtering model. The Pass@1 result for BFS for No Filtering was 44.8. We observe 3D-Prover outperforming both the No Filtering and Top-$K$ baselines, with a notable improvement over more attempts for the Critic Guided search model.

| | Top-$K$ ($K = 8$) | 3D-Prover ($K = 8$) | Gain |
|---|---|---|---|
| *InternLM2.5-StepProver (BFS)* | | | |
| Pass@1 | 44.3 (44.1, 44.5) | **45.7 (45.3, 46.1)** | +3.2% |
| Pass@2 | 44.9 | **47.3** | +5.3% |
| | No Filtering ($K = 64$) | 3D-Prover ($K = 8$) | Gain |
| *InternLM2.5-StepProver (Critic Guided)* | | | |
| Pass@1 | 43.7 (42.4, 44.5) | **45.7 (44.5, 47.0)** | +4.6% |
| Pass@6 | 49.0 | **53.1** | +8.4% |

(Table 3) demonstrate this, with the improvements from filtering growing over more attempts. This indicates a better variety of paths being explored, as each different attempt is more likely to take a more diverse approach. Finding deep proofs is a significant challenge Polu et al. (2022), with the search tree growing exponentially with proof depth. The large improvements from 3D-Prover for deeper search is a step towards addressing this.

Tree search should be considered an augmentation of the base model, with improvements generally much smaller than those found by improving the generator itself. This is unsurprising, as the generator forms the base set of candidates for the search to explore. Improved search algorithms do, however, have the advantage of being applicable to different base models, which is important given the rapid advance of new and better generators. For example, DeepSeek-Prover-V1.5 obtains around 2–4% relative improvements in proof success over miniF2F-test with its novel tree search algorithm, compared to no search. In comparison, improving their base model yields a ~36% relative improvement (Figure 5 and Table 1 in (Xin et al., 2024)). Similarly, Table 1 from Polu et al. (2022) shows their search approach yielding 0.04-5.7% relative improvements for miniF2F-valid, with ~40,000 GPU hours required for their best results. We were able to find our improvements with significantly less resources, training our transition model on only a single attempt per proof.

We emphasise that these results were obtained without any hyperparameter tuning, only using the representations as diversity features and model logits as quality scores. We present ablation studies looking closer at these hyperparameters, however a comprehensive sweep is prohibitively expensive (Appendix I). Despite this, we were able to obtain our improvements without tuning, demonstrating the effectiveness of our approach. For completeness, Appendix C details the Pass@1 performance over the hyperparameter configurations we tested for our ablations. We also highlight that the miniF2F-test ReProver results were obtained by training with transitions from miniF2F-valid, showing that 3D-Prover remains effective for proofs out of distribution. The results on miniF2F-valid represent the more common online scenario, with previous attempts on the same dataset being used to improve performance (see, for example, Lample et al. (2022); Polu et al. (2022)). We also note that our approach is lightweight with minimal overhead, as we detail in Appendix G.

### 3.1.2 Ablation Study

**Effect of the Transition Model** To demonstrate the utility of our tactic representations, we compare to an ablated 3D-Prover where the transition model Encoder is replaced by an Autoencoder of the same size. The Autoencoder is trained to reconstruct the original tactic, and therefore generates representations which reflect only the syntax of the tactic. This tests our hypothesis that semantically aware tactic representations are useful for proofs, justifying the inclusion of the transition model. From Table 4, the performance of 3D-Prover with the transition model embeddings is indeed superior to that of the Autoencoder across all values of $K$. This shows that selecting for diversity with respect to the predicted semantics, rather than the syntax, leads to a direct improvement in proof performance.

Table 4: Percentage of proofs found after one attempt (Pass@1) on miniF2F-valid, comparing 3D-Prover with a Transition Model Encoder to an Autoencoder trained to reconstruct the original tactics. We see that 3D-Prover with the Transition Model gives a clear improvement in proof success over the Autoencoder, demonstrating the utility of our representation architecture in Section 2.

|  | $K = 8$ | $K = 16$ | $K = 32$ |
|---|---|---|---|
| Autoencoder | 23.0 | 27.9 | 27.0 |
| 3D-Prover | **25.0** | **29.1** | **28.7** |

We have demonstrated that 3D-Prover improves proof success rate without any hyperparameter tuning, with a fixed $\lambda_s = \lambda_\tau = 0$, $\theta = 1$. We now examine whether we can use 3D-Prover to direct search to optimise secondary objectives, namely the execution time, tactic success rate and the diversity of environment response.

**Success Rate** We observe from Table 5 3D-Prover significantly improves the success rate of chosen tactics. As $K$ decreases, this improvement increases in magnitude, reflecting the heightened influence of the filtering model. We see that this improvement increases with the error term $\lambda_s$, which scales the quality scores of tactics by their predicted probability of success, as was intended.

Table 5: Tactic success rate per node for miniF2F-valid (Mean $\pm$ Standard Error), where $\lambda_s$ controls the error weight of quality score in 3D-Prover. No filtering gives $27.7\% \pm 0.2\%$. We see that 3D-Prover leads to fewer errors on average, which can be controlled by increasing $\lambda_s$.

| $K$ | Top-$K$ | Random | 3D-Prover ($\lambda_s = 0.1$) | 3D-Prover ($\lambda_s = 0.5$) |
|---|---|---|---|---|
| 8 | $39.0 \pm 0.1$ | $33.4 \pm 0.1$ | $43.3 \pm 0.1$ | $\mathbf{56.5 \pm 0.1}$ |
| 16 | $39.0 \pm 0.1$ | $30.9 \pm 0.1$ | $40.0 \pm 0.1$ | $\mathbf{51.7 \pm 0.1}$ |
| 32 | $35.0 \pm 0.2$ | $29.7 \pm 0.1$ | $35.7 \pm 0.1$ | $\mathbf{41.7 \pm 0.1}$ |

**Diversity** To measure diversity, we examine the percentage of successful tactics which result in a new proof path. We restrict to successful tactics to account for the discrepancy in success rate between approaches. We observe 3D-Prover gives more unique subgoals per successful tactic, which is noteworthy given the higher rate of successful tactics from 3D-Prover overall (Table 5). As intended, increasing $\theta$ gives further improvements under these metrics. This demonstrates that our approach is effective at avoiding redundant tactics, instead selecting tactics which yield more unique proof paths. Appendix F provides additional analysis, further supporting our claim of improved diversity.

Table 6: Percentage of successful tactics per node resulting in unique subgoal(s) over miniF2F-valid (Mean $\pm$ Standard Error). No filtering gives $67.8\% \pm 0.3\%$. We observe 3D-Prover results in more unique subgoals per tactic, leading to a more diverse set of proof paths, with larger $\theta$ controlling this.

| $K$ | Top-$K$ | Random | 3D-Prover ($\theta = 1$) | 3D-Prover ($\theta = 4$) |
|---|---|---|---|---|
| 8 | $85.3 \pm 0.1$ | $89.9 \pm 0.1$ | $90.1 \pm 0.1$ | $\mathbf{91.1 \pm 0.1}$ |
| 16 | $77.5 \pm 0.1$ | $84.1 \pm 0.1$ | $84.9 \pm 0.1$ | $\mathbf{85.5 \pm 0.1}$ |
| 32 | $72.3 \pm 0.2$ | $76.3 \pm 0.2$ | $76.9 \pm 0.2$ | $\mathbf{77.5 \pm 0.2}$ |

**Execution Time** Table 7 shows the execution time for tactics over miniF2F-valid transitions. Again we see that 3D-Prover outperforms the baselines, with the improvement increasing with more filtering. Increasing the time weight $\lambda_\tau$ results in further reductions to the average execution time, demonstrating the accuracy of the predictions, and that they can directly result in faster tactics when filtering. It has been noted that preferring faster tactics can prevent the excessive application of powerful automation tactics such as `simp` (Lample et al., 2022, Appendix E). As these generally take longer to run, using faster tactics can help models learn underlying proof arguments which are often hidden by the automation. It can also greatly reduce the number of timeout errors.

Table 7: Tactic execution time in milliseconds over miniF2F-valid proof attempts (Mean $\pm$ Standard Error). No filtering resulted in $232\pm 0.9$ milliseconds. $\lambda_\tau$ controls the time weighting of the quality score in 3D-Prover. 3D-Prover selects faster tactics on average, with larger $\lambda_\tau$ magnifying this.

| $K$ | Top-$K$ | Random | 3D-Prover ($\lambda_\tau = 0.1$) | 3D-Prover ($\lambda_\tau = 1.0$) |
|---|---|---|---|---|
| 8 | $206 \pm 0.8$ | $198 \pm 0.9$ | $155 \pm 0.5$ | $\mathbf{136 \pm 0.5}$ |
| 16 | $220 \pm 0.8$ | $218 \pm 0.9$ | $176 \pm 0.6$ | $\mathbf{152 \pm 0.5}$ |
| 32 | $224 \pm 0.8$ | $215 \pm 0.8$ | $191 \pm 0.7$ | $\mathbf{181 \pm 0.6}$ |

## 4 Conclusion

**Limitations** Our main limitation was the scale of experiments we were able to run, with other results often requiring thousands of hours of train time using hundreds of provers and larger models (Lample et al., 2022; Polu et al., 2022; Xin et al., 2024). Given the large computational cost of evaluations (as we outline in Appendix I), we were only able to test InternLM2.5 up to Pass@2 for BFS and Pass@6 for Critic Guided, and ReProver up to Pass@4 on miniF2F-test, with a hyperparameter analysis of 15 runs on miniF2F-valid (Table C). We could only evaluate over the large LeanDojo benchmark (Appendix D) for single run with each approach.

**Summary** We demonstrate the feasibility of learning proof system dynamics, where we generate tactic representations reflecting the response of the proof environment. We then leverage these with 3D-Prover, which filters candidate tactics to diverse and high quality subsets based on their likely outcome. We evaluate 3D-Prover by augmenting popular proving LLMs on the standard miniF2F and LeanDojo benchmarks, where we find an improvement in the overall proof success rate, particularly for deeper searches. Our ablation studies confirm the utility of our tactic representations, enabling the selection of tactics with improved success rates, diversity, and/or execution time. By effectively pruning the search space, 3D-Prover is a step towards enabling deeper automated proofs.

## 5 Acknowledgements

We acknowledge Defence Science and Technology Group (DSTG) for their support in this project.

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

## A    Proof Search Setup

We first define the space of goals $\mathcal{S}$, tactics $\mathcal{T}$ and failures $\mathcal{F}$. For our purposes, these all contain arbitrary strings, with the goal being a formal proposition, the tactic a command and the failure an error message. We then define the output space as $\mathcal{O} := \mathcal{P}(\mathcal{S}) \cup \mathcal{F}$. A *proof tree* is a DAG $G = (V, E)$ where $V \subset \mathcal{S}$ is the set of goals and $E$ the edges between them. A *proof attempt* for a goal $g_0$ first initialises the proof tree with $V = \{g_0\}, E = \emptyset$. The *search policy* $\pi_S : G \times V \to \mathbb{R}^+$ is a distribution over goals given a proof tree, being used to select a goal $g$ to expand. The *tactic policy* $\pi_T : \mathcal{S} \times \mathcal{T} \to \mathbb{R}^+$ is a distribution over tactics given a goal, where $N \in \mathbb{N}$ tactics are sampled to give tactics $\{t_i\}_{i=1}^N \subset \mathcal{T}$. The goal, tactic pairs $(g, t_i)$ are then passed to the environment $\mathcal{E} : \mathcal{S} \times \mathcal{T} \to \mathcal{O}$. For each pair, after $\tau_i \in \mathbb{R}$ seconds, it returns either a new set of goals $g_i' \subset \mathcal{S}$ or an error, $e_i \in \mathcal{F}$. We define this response as the *output* $o_i \in \mathcal{O}$. We further define the *status* $s_i \in \{0, 1\}$ as 0 if $o_i \in \mathcal{F}$, 1 if $o_i \in \mathcal{P}(\mathcal{S})$ and the *transition* as the tuple $(g, t_i, s_i, \tau_i, o_i)$. The proof tree is then updated with $G = G \cup g_i'$ for all $g_i'$, and the associated transitions are added as edges to $E$. This is repeated until a *proof* is found, or a budget is exhausted. A proof of $g$ is found when $\mathcal{E}(g, t_i) = \emptyset$ for any $t_i$, or if all $\{g_i'\}$ are proven for $\mathcal{E}(g, t_i) = \{g_i'\} \subset \mathcal{S}$. The result of a proof attempt is then the set of transitions $\{(g_k, t_{ki}, s_{ki}, \tau_{ki}, o_{ki})\}$ for all selected goals $g_k$ and their expanded tactics $t_i$. For simplicity, we drop the indices to denote the set of transitions as $\{(g, t, s, \tau, o)\}$.

## B    Pass@k

Table 8 summarises the Pass@4 results for ReProver over miniF2F-test, which is the number of proofs found at least once over four attempts, with Table 9 showing the Pass@$k$ up to $k = 4$. We compare

Table 8: Percentage of proofs found after four attempts (Pass@4) with ReProver on miniF2F-test, with $K$ tactics selected per node.

| $K$ | Random | 3D-Prover | Gain |
|---|---|---|---|
| 8 | 25.7 | **28.6** | +11.3% |
| 16 | 30.2 | **31.0** | +2.6% |
| 32 | **29.8** | **29.8** | +0.0% |

Table 9: Pass@$k$ rates for proof attempts on miniF2F-test

| | 3D-Prover | | | Random | | |
|---|---|---|---|---|---|---|
| Pass@$k$ | $K = 8$ | $K = 16$ | $K = 32$ | $K = 8$ | $K = 16$ | $K = 32$ |
| 1 | 24.9 | 27.8 | 28.6 | 18.0 | 21.2 | 28.1 |
| 2 | 26.1 | 29.4 | 29.0 | 22.9 | 28.6 | 29.0 |
| 3 | 26.5 | 29.8 | 29.8 | 24.9 | 29.4 | 29.8 |
| 4 | 28.6 | 31.0 | 29.8 | 25.7 | 30.2 | 29.8 |

3D-Prover to the Random baseline, taking the same four runs from Table 2, where $\lambda_s = \lambda_\tau = 0$, $\theta = 1$. With Top-$K$ being deterministic, the Pass@$k$ rate is the same as the Pass@1 rate. Given several attempts, $K = 16$ appears to provide a good tradeoff between breadth and depth, performing the best overall. 3D-Prover maintains a large improvement for $K = 8$, with a modest improvement for $K = 16$.

As discussed by Chen et al. (2021), the Pass@$k$ metric favours exploratory approaches as $k$ increases, at the cost of lower performance for smaller $k$. This is because, over many attempts, a highly exploratory approach is more likely to find at least one proof of a given goal, even though it may find fewer proofs in a single attempt than a more exploitative approach. Further discussion in Lample et al. (2022) finds that randomly sampling search parameters also improves Pass@$k$. With Pass@$k$ being expensive to estimate, we fix our parameters over the four runs to give a more accurate estimate of Pass@1. Given this, a large scale experiment sampling these hyperparameters could lead to improved Pass@k results, as Lample et al. (2022) show for their HTPS approach.

## C  Proof success rate over hyperparameters

Table 10: Pass@1 results on miniF2F-valid, over different hyperparameter configurations for 3D-Prover with ReProver.

| | $(\lambda_s, \lambda_\tau, \theta)$ | | | | |
|---|---|---|---|---|---|
| $K$ | (0.0, 0.0, 1.0) | (0.1, 0.1, 1.0) | (0.5, 0.1, 1.0) | (0.1, 1.0, 1.0) | (0.1, 0.1, 4.0) |
| 8 | 25.0 | 25.0 | **25.8** | 22.5 | 23.8 |
| 16 | **29.1** | 28.7 | 27.9 | 27.0 | 26.6 |
| 32 | **28.7** | 28.3 | **28.7** | 27.9 | 27.0 |

Table 10 shows the Pass@1 results on miniF2F-valid for 3D-Prover for our limited hyperparameter sweep. These results suggest that a lower time weight $\lambda_\tau$ leads to better proving results. The diversity parameter $\theta$ hinders performance for the larger value, consistent with what was observed by Chen et al. (2021), where they observe a tradeoff between exploration and Pass@1. Although these parameters may not improve Pass@1, different proofs may favour different configurations, with some requiring e.g. more depth or exploration than others. As discussed above, a higher Pass@$k$ can usually be obtained by sampling a wide set of these parameters. For the set of hyperparameters we tested here, we found a cumulative proof rate (or Pass@15) of 32.8% on miniF2F-valid.

# D    Evaluation on LeanDojo Benchmark

We ran an additional experiment on the LeanDojo Novel Premises Yang et al. (2023) benchmark testing 3D-Prover on a larger dataset. This dataset has 2000 evaluation proofs in comparison to the 244 from miniF2F-valid and miniF2F-test, allowing us to evaluate the performance of 3D-Prover on a larger scale.

We trained a transition model from a single ReProver attempt on LeanDojo Novel Premises, before evaluating 3D-Prover using ReProver with the methodology in Section 3. We set $K$=32 for 3D-Prover, and compare to the model with No Filtering (i.e. $K$=64), and Top-$K$=32. We further examine the distribution of proof lengths found from this experiment. To account for different proofs of the same goal, we adjust proof lengths to be the shortest found from any attempt ( e.g. if 3D-Prover finds a proof of length 10, which was found in 3 steps by No Filtering, we count it as length 3). Hence, all proof lengths reported are the shortest found by any method. We report the number of proofs found by each approach, organised by the proof length in Table 11.

Table 11: Number of Proofs found on LeanDojo Novel Premises, sorted by proof length.

| Proof Length | 3D-Prover ($K = 32$) | Top-$K$ ($K = 32$) | No Filtering ($K = 64$) |
|---|---|---|---|
| 1 | 236 | 233 | **237** |
| 2 | 167 | 162 | **174** |
| 3 | **134** | 126 | 131 |
| 4 | **60** | **60** | 54 |
| 5 | **40** | 39 | 24 |
| 6 | **7** | 6 | 2 |
| 7 | **2** | 0 | 0 |
| Total | **646** | 626 | 622 |
| Pass@1 | **32.3%** | 31.3% | 31.1% |

3D-Prover obtains a relative improvement of 3.2% over Top-$K$, and a 3.9% relative improvement over No Filtering in terms of the number of proofs found, comparable to the same performance gain for $K = 32$ found in miniF2F-valid (Table 2). We see that 3D-Prover finds deeper proofs, while maintaining a high proof success rate for shallower proofs, unlike Top-$K$. The No Filtering approach, as expected, finds the most shallow proofs, however quickly drops off in performance for deeper proofs. We also note that 3D-Prover found the 2 longest proofs of length 7, with neither baseline finding any.

# E    Embedding Discussion

**Embedding Comparison**   We now investigate whether the transition model (Figure 2) captures tactic semantics rather than syntax in its tactic embeddings. To test this, we examine the cosine similarity of tactic embeddings which lead to unique subgoals. Figure 4 takes an example node, examining all tactics which lead to a unique subgoal. The upper value displays the cosine similarity given by the transition model, while the lower value displays that given by the Autoencoder in Section 3.1.2. We observe that in most cases, the similarity given by the transition model is much lower than that given by the Autoencoder, which is only considering the syntax of the tactic. For example, the similarity between tactic 3 and 4 is very high for the Autoencoder, given the similar syntax between the two as they use the same lemma. Despite this similar syntax, the transition model embeddings show a high degree of dissimilarity, reflecting the different outcome they have on the environment. We present additional examples in the supplementary code. To generalise beyond these examples, we ran this comparison over the tactic embeddings which lead to unique subgoals for all 244 root nodes in minF2F-valid. Figure 5 shows the distribution of the average cosine similarity for each node, for both the transition model and the Autoencoder. The average cosine similarity for the transition model embeddings was 0.44 while the Autoencoder gave 0.57. While this comparison does not account for similarity between the unique subgoals, it is still clear that the transition model

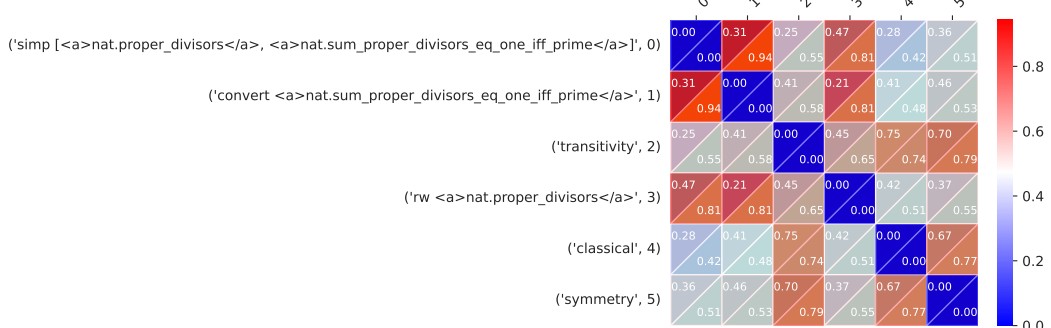

Figure 4: Cosine similarity between tactic embeddings resulting in unique subgoals, for a sample root node in miniF2F-valid. The top value gives the similarity for embeddings from 3D-Prover, while the bottom gives the similarity for embeddings from an Autoencoder. We see that 3D-Prover better separates these semantically distinct tactics, in comparison to the Autoencoder, which only separates based on their syntax.

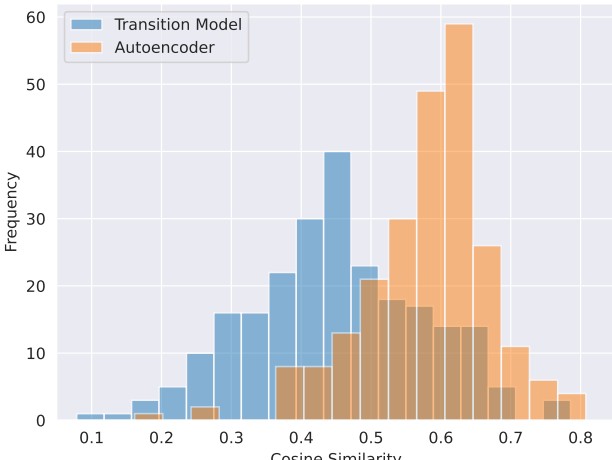

Figure 5: Distribution of cosine similarity for tactic embeddings resulting in unique subgoals, averaged over root nodes in miniF2F-valid. We see that 3D-Prover gives embeddings which better separate these semantically distinct tactics, in comparison the the syntax focused embeddings of the Autoencoder.

embeddings better separate unique tactics than Autoencoder embeddings which are based on syntax alone. The result of this is a higher likelihood of 3D-Prover selecting tactics which give unique subgoals, which as we show in Section 3.1.2, results in the transition model outperforming the Autoencoder for proof discovery.

**Embedding Objective** As outlined in Section 2, we train our embeddings to be reflective of the tactic semantics across all three components of Status, Time and Output. Hence 3D-Prover, which selects diverse embeddings, may lead to tactics predicted to have errors, where the errors are diverse in terms of their predicted message. The hyperparameter $\lambda_s$ can alleviate this by weighting the scores based on their likelihood of success. From our experiments (Table 10), there is not necessarily a benefit to Pass@1 by filtering out strongly based on the predicted error likelihood. To speculate, the error prediction, although quite good, is imperfect with false negatives (Table 1). This can lead to potentially useful tactics being ignored if the error prediction is overly trusted, even though there is a higher tactic success rate overall as in Table 5. Given these prediction errors, it may be the case that selecting goals which are predicted to lead to (diverse) errors may be preferable, given the possibility they result in successful new subgoals. These subgoals may be be quite different from those previously selected, as they are mispredicted, so are clearly outside the space of tactics

where the transition model is confident about the outcome. Further analysis could be worthwhile to investigate this. An embedding architecture trained only on successful tactics could be used, however given the high error rate of tactics, this would ignore a large proportion of the transition data.

# F    Additional Diversity Analysis

To further examine diversity, we first look at the percentage of unique environment responses to tactics executed per node, including responses with unique errors (Table 12), using the same ReProver setup in 3. As it is difficult to select tactics guaranteed to be successful (see Table 5), we would expect a good exploratory policy to select tactics which result in more varied outputs (both errors and successes alike), so as to better explore the space. We also examine the degree of cosine similarity across unique subgoals (Table 13), using a simple text embedding model (`all-MiniLM-L6-v2`). This more precisely quantifies the diversity in the contents of the subgoals. We do this to account for simple changes (such as variable renaming), which would not be differentiated under a simple check for uniqueness, and so allows us to examine how varied the contents of the resulting subgoals are.

In both cases, we see that 3D-Prover results in more diverse responses. As intended, increasing $\theta$ results in further improvements to diversity under both metrics. The increased diversity in subgoal content (Table 13) is strengthened by the fact that 3D-Prover also gives more unique subgoals on average (Table 6), suggesting that our approach yields more unique subgoals, and that these subgoals are more varied in their contents.

Table 12: Percentage of unique environment responses per node in miniF2F-valid (Mean $\pm$ Standard Error). Unique defines either syntactically distinct error messages or responses including at least one previously unseen subgoal. No filtering results in $63.3\% \pm 0.2\%$. We see that 3D-Prover gives a higher diversity of environment responses, increasing with the diversity parameter $\theta$.

| | | | 3D-Prover | |
| $K$ | Top-$K$ | Random | $\theta = 1$ | $\theta = 4$ |
|---|---|---|---|---|
| 8 | $83.9 \pm 0.1$ | $88.6 \pm 0.1$ | $90.8 \pm 0.0$ | $\mathbf{91.7 \pm 0.0}$ |
| 16 | $77.5 \pm 0.1$ | $81.4 \pm 0.1$ | $85.9 \pm 0.1$ | $\mathbf{86.6 \pm 0.1}$ |
| 32 | $71.1 \pm 0.1$ | $72.7 \pm 0.1$ | $77.6 \pm 0.1$ | $\mathbf{78.1 \pm 0.1}$ |

Table 13: Average cosine similarity between subgoal embeddings for ReProver over miniF2F-valid (Mean $\pm$ Standard Error). We observe consistently lower similarity among subgoals generated from 3D-Prover, increasing with the diversity parameter $\theta$.

| | | | 3D-Prover | |
| $K$ | Top-$K$ | Random | $\theta = 1$ | $\theta = 4$ |
|---|---|---|---|---|
| 8 | $0.939 \pm 0.000$ | $0.945 \pm 0.000$ | $0.931 \pm 0.000$ | $\mathbf{0.930 \pm 0.000}$ |
| 16 | $0.916 \pm 0.000$ | $0.926 \pm 0.000$ | $0.910 \pm 0.000$ | $\mathbf{0.905 \pm 0.000}$ |
| 32 | $0.904 \pm 0.000$ | $0.909 \pm 0.001$ | $0.900 \pm 0.001$ | $\mathbf{0.896 \pm 0.001}$ |

# G    Computational Overhead

3D-Prover adds a constant but minimal time and memory overhead, the majority of which is in generating embeddings for the candidate tactics. Taking our first run for 3D-Prover with InternLM, we found the filtering time over nodes to be $0.07$s with a standard deviation of $0.02$s. In comparison, the average tactic generation time was $7.5$s with a standard deviation of $4$s. This gives us a time overhead of approximately $0.9\%$. The memory overhead was approximately 3GB of VRAM, while the tactic model took 44GB of VRAM, giving approximately $7\%$ memory overhead.

We also note that the average time to execute a tactic from the No Filtering model was approximately $0.13$s (with $0.12$ standard deviation). With up to 64 candidate tactics per node, the filtering time

of 0.07 seconds is around half the average time to execute a single tactic. Given the improvements in success rate (Table 5), our filtering model therefore gives an effective way to reduce the total computational resources by preventing the wasteful execution of erroneous tactics. To further support this, we observed the average total time for a proof search attempt with 3D-Prover with InternLM ($K = 8$) to be 993.8 seconds (1761.5 Standard Deviation), while the total time for Top-$K = 8$ was 1116.8 seconds (2126.0 Standard Deviation).

## H    Number of tactic candidates for InternLM

As noted in 3, our experiments with InternLM2.5-Step-Prover use sampling rather than beam search for tactic generation, as done in Wu et al. (2024). As a result, there is no guarantee there will be 64 unique tactics, with many samples being identical. As a result, we sample 128 initial tactics, and take the total unique candidates as our ground set (up to 64 total). We plot the distribution of unique tactic candidates per node in Figure 6. This informed our choice of $K = 8$ for our experiments, as higher $K$ would give minimal filtering in comparison to a beam search approach.

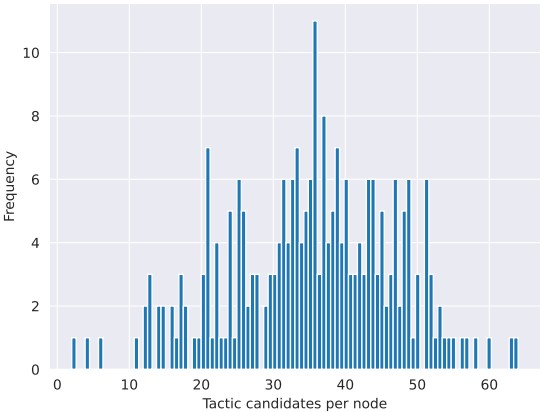

Figure 6: Distribution of unique tactic candidates per node for InternLM2.5-Step-Prover over miniF2F-test ($\mu = 35.4, \sigma = 11.5$).

## I    Computational Resources and Usage

For traning our transition models, we used a singe RTX4090 GPU with an Intel i9 13900k processor. For evaluating, we used two internal machines each with two RTXA6000 GPUs, and a Intel Xeon W-2223. For each evaluation experiment in 3, we assigned a single RTX A6000 GPU which contained both the tactic generator and transition model, which served 2 CPU provers which request tactics and evaluate in the Lean environment.

For each transition model, training for 2 epochs took approximately 2 days for each run, giving a total of 10 days of 4090 training for our results in 2.2. Each evaluation run for ReProver over miniF2F took around 12h, while each evaluation run for InternLM2.5-Step-Prover around 2 days. Counting the number of runs for all baselines and 3D-Prover, we have 28 runs for ReProver with miniF2F-test, 22 for miniF2F-valid and 17 runs for InternLM. The additional result over the large LeanDojo dataset in D took around 5 days per run, with 7 runs total. We therefore estimate the total evaluation time (for a single machine with 2 RTX6000s and a Xeon W-223) to be 94 days, which was around 47 days with our 2 machines. The full research project included additional compute and experiments, as different architectures and approaches were prototyped, however we do not have an estimate for the amount.

