# OpenReview forum: "3D-Prover: Diversity Driven Theorem Proving With Determinantal Point Processes"
_NeurIPS.cc/2025/Conference — NeurIPS 2025 poster_

### Official Review · Reviewer_SA37 · 2025-06-20

**Clarity:** 4
**Significance:** 3
**Originality:** 4
**Rating:** 5
**Confidence:** 4

**Summary:**

This paper introduces a novel search algorithm for guiding LLM-based automated (interactive) theorem proving. The core idea is to train a small model to predict the "dynamics" of the theorem proving environment, and to then use this model as a signal from which to select a set of diverse and high-quality tactics at each timestep. In order to turn the signal into a distribution over the tactics which incorporates both notions, the authors leverage Deterministic Point Processes (DPP). The experiments are somewhat limited in scope, but suggest this method can lead to small (but meaningful, if statistically significant) improvements over existing methods. As the authors point out, while improving search is unlikely to ever yield as large of an improvement as improving the policy model, progress in this area is still meaningful as it is complementary to works seeking to improve the base model.

**Questions:**

Suggestions:
- In the introduction there is a typo: "It applications"
- On page, "columns b_i" should likely be "rows b_i" (or you have accidentally transposed your matrix dimensions)

Questions:
- In table 3, you state that you report the mean += one std dev over 2 runs for pass@1. Am I understanding correctly that you literally computed the arithmetic mean and sample std dev over 2 samples? That is not a meaningful metric, if so. If your sample size is small it would be much better to report (min, max) ranges, or a confidence interval which does not rely on asymptotic normality (eg based on Hoeffding's inequality).
- In general, do you have any supporting (statistical) evidence that suggests your findings are even statistically significant? I am sympathetic to the difficulties of doing research in this field without access to large computational resources, but as it stands I am not entirely convinced that the single-digit improvements reported downstream are significant.

**Ethical Concerns:**

["NO or VERY MINOR ethics concerns only"]

**Final Justification:**

This paper is not without its flaws. The experiments could really use some fleshing out in order to give a more confident account of when and how this method is applicable; in particular, I agree with Bo1h that the sensitivity of the method to the additional hyperparameters is understudied, which I not believe was resolved completely during the rebuttal. Other reviewers have also rightly questioned the impact of this method, given the marginal gains.

However, the paper does a good job of explicitly calling out and addressing these limitations, which I believe to be commendable, and with the changes made to the presentation of the results I am confident that the overall methodology is both novel and valid. I do not think the fact that the gains are not very large is sufficient grounds to reject the paper, since (as pointed out by the authors) improvements in the search method complement other developments in the literature, such as having a stronger base model.

In conclusion, I find this paper well scoped and written. I believe the community will find the methodology interesting, and that it may prove valuable in practice. I therefore recommend the paper for acceptance.

**Limitations:**

yes

**Quality:**

4

**Strengths And Weaknesses:**

Strengths:
- The idea is, as far as I know, novel. Leveraging DPPs in particular is a fun and interesting approach to balanced diversity and quality.
- The paper is well-written, with clear figures and methodology and no unnecessary formalizations (that is: the math that is in the paper serves its purpose).
- The limitations of the study, including limited (but non-zero) impact on downstream performance and the limited scope of the experiments, are both acknowledged and discussed extensively in the paper.
- I appreciate that the authors quantitatively evaluated the overhead that this search procedure introduces. In particular, a priori I expected that running the pipeline on each generated tactic would exceed the time it takes to simply execute that tactic, however this appears not to be the case as per Appendix G. This makes me much more optimistic about the method's practical relevance.
- Work in this area is complementary to a lot of the other stuff that is happening in NLP right now, which from the point of view of interactive theorem proving largely boils down to improving the base (policy) model.

Weaknesses:
- While it is good that the authors acknowledge that a lack of computational resources casts some doubt on the signifance and generalizability of their findings, it is hard to quantify how big this effect is. As a result, future work would need to confirm that this methodology indeed works as well as these initial findings would suggest.

---

> ### Author Rebuttal · Authors · 2025-07-30
>
> We are grateful to Reviewer SA37 for their thorough and encouraging review, and we appreciate their recognition of our work's novelty and clarity, particularly our "fun and interesting" use of DPPs and the quantitative analysis of computational overhead. We are also grateful for their typo suggestions, which we will incorporate into the revision.
>
> ## W1 / Q1 / Q2
> - *While it is good that the authors acknowledge that a lack of computational resources casts some doubt on the signifance and generalizability of their findings, it is hard to quantify how big this effect is. As a result, future work would need to confirm that this methodology indeed works as well as these initial findings would suggest.*
> -*In table 3, you state that you report the mean += one std dev over 2 runs for pass@1. Am I understanding correctly that you literally computed the arithmetic mean and sample std dev over 2 samples? That is not a meaningful metric, if so. If your sample size is small it would be much better to report (min, max) ranges, or a confidence interval which does not rely on asymptotic normality (eg based on Hoeffding's inequality).*
> -*In general, do you have any supporting (statistical) evidence that suggests your findings are even statistically significant? I am sympathetic to the difficulties of doing research in this field without access to large computational resources, but as it stands I am not entirely convinced that the single-digit improvements reported downstream are significant.*
>
> We very much appreciate your thoughtful comments and your recognition of the inherent difficulties in conducting research in Neural Theorem Proving (NTP) without access to extensive computational resources.
>
> As we discuss in our response to Q3 from Reviewer Bo1h, we have been running additional experiments since submission to obtain larger Pass@k values (and thereby better estimates for Pass@1). **Our results here also represent a larger improvement**. We are actively working to expand our experimental runs to enable more robust statistical analysis in future work.
>
> Regarding your specific comment on Table 3 and the reporting of mean ± one std dev over 2 runs for pass@1: We fully agree that computing the arithmetic mean and sample standard deviation over only two samples is not a statistically meaningful metric. Given the small sample sizes, we acknowledge that this reporting could be improved. For greater transparency, we will revise Table 2 and 3 to report the mean pass@1 value, as well as the maximum and minimum values. The results are as below:
>
> ## Pass@1 adjusted (average, max, min) for Table 2
>
> | K | Random (Avg, Max, Min) | 3D-Prover (Avg, Max, Min) |
> |---|------------------------|---------------------------|
> | 8 | (19.0, 19.6, 18.4) | **(24.4, 24.9, 23.7)** |
> | 16 | (25.4, 25.7, 24.5) | **(27.3, 27.8, 26.9)** |
> | 32 | (27.4, 28.2, 26.9) | **(28.2, 28.6, 27.3)** |
>
> ## Pass@1 adjusted (average, max, min) for Table 3
>
> | Approach | Average | Max | Min |
> |:---|---:|---:|---:|
> | 3D-Prover (K=8) | **45.7** | **46.1** | **45.3**|
> | No Filtering | 44.8 | 45.7 | 43.7 |
> | Top-8 | 44.3 | 44.5 | 44.1 |
>
> The broader point about the lack of statistical significance for Pass@k values is highly relevant and reflects a common practice within the NTP community. Pass@k values in the current literature are often reported as point estimates from a single value, often without explicit statistical significance measures such as confidence intervals or p-values.
>
> For example, the highly influential [1] report pass@k values without associated statistical significance. Similarly, the recent ICLR25 CARTS [2] reports pass@l rates over the miniF2F-test benchmark explicitly noting that their paper "does not explicitly report on statistical significance for the experimental results".
>
> This practice is largely a direct consequence of the prohibitive computational cost associated with NTP experiments. The process of generating and verifying formal proofs demands considerable resources, given the integration of large LLMs, costly search algorithms and CPU intensive verification. For instance, [2] explicitly states that "due to computational cost limitation, we only compared the results for one single tree search attempt". This inhibits the multiple repetitions necessary for robust confidence intervals or p-values.
>
> We want to assure you that we are committed to statistical rigor where feasible. To address your concern about statistical significance, we highlight that our hyperparameter ablations (Tables 5, 6, and 7) do show statistically significant improvements. Furthermore, the reduction in average proof time presented in Appendix G was significant (p < 0.05), which provides statistical evidence for speeding up proof attempts.
>
>
> [1]:
> Lample et al. (2022). HyperTree Proof Search for Neural Theorem Proving. https://proceedings.neurips.cc/paper_files/paper/2022/file/a8901c5e85fb8e1823bbf0f755053672-Paper-Conference.pdf
>
> [2]:
> Yang et al. (2025). CARTS: Advancing Neural Theorem Proving with Diversified Tactic Calibration and Bias-Resistant Tree Search. https://openreview.net/pdf?id=VQwI055flA

---

> > ### Comment · Reviewer_SA37 · 2025-08-03
> > **Response to Rebuttal**
> >
> > I thank the authors for their reply to my and my fellow reviewers' comments.
> >
> > Presenting the main results as (average, max, min) is a significant improvement. While this is not a complete substitute for statistical significance testing, it does lend more credibility to the performance gains, as the baselines' max rarely meet 3D-Prover's min. In addition, I appreciate that the authors have shown good faith in this regard by carrying out statistical significance testing when possible (hyperparam sweeps and runtime improvements).
> >
> > Having read the other reviews (and the responses thereof), I note that there is a general concern about the significance of the results. My opinion remains aligned with that of the author: improvements in the search method itself are unlikely to yield the large performance gaps we have become accustomed to from improvements made in the base models, but this does not mean that these results are not useful for the field. In particular, since the search strategy is orthogonal to the policy, these small gains may still find broad applicability.
> >
> > As to the other comments, I am most sympathetic to Bo1h's concerns about the lack of a more extensive hyperparameter sweep. While I agree with the authors that it is a good sign that they saw performance improvements without having to carefully tune hyperparameters, Table 10 in Appendix C does suggest that the method may nonetheless actually be quite sensitive to these values. Furthermore, if $\lambda_s = \lambda_\tau  = 0$ performs so well, it begs the question of why additional machinery of predicting $(s, \tau)$ is even needed. However, section 3.1.2 does shed some light on how these hyperparameters may be used to elicit particular behaviour from the model at test time, with the only cost being training the very small Predictor network. Ideally it would have been good to see a concrete example of this contributing positively to downstream performance, but I am sympathetic to the fact that benchmarks in this domain are quite limited, and that pass@1 is a crude metric for mathematical theorem proving.
> >
> > Overall, I am satisfied with the responses to the concerns raised by the other reviewers. I will wait with officially adjusting my score in case there is additional discussion with the other reviewers that brings new concerns to light, but I currently intend to raise my score to 5 - Accept. I have no further requests for the authors.

---

> > > ### Author Response · Authors · 2025-08-07
> > >
> > > We thank reviewer SA37 for their further support of our work and active engagement in the discussion. We especially appreciate their suggestions, which we agree have improved the clarity and credibility of our results.

---

### Official Review · Reviewer_YkZF · 2025-07-03

**Clarity:** 2
**Significance:** 3
**Originality:** 3
**Rating:** 4
**Confidence:** 3

**Summary:**

The paper addresses a critical issue in the intersection of LLMs and Lean: the semantic redundancy among multiple strategies generated by LLMs, which significantly increases unnecessary computational overhead. To tackle this, the authors propose 3D-Prover, a strategy selection algorithm for automated theorem proving. Its objective is to select a subset of candidate strategies that are both high-quality and diverse, thereby improving proof efficiency and success rates.

**Questions:**

1.If the predictor ( P ) does not predict the execution time, how much would it affect the performance of the algorithm?
2.Did the encoder, predictor, and decoder in the paper reuse the internal architecture of the LLM, or were they additional components introduced by the authors?
3.My understanding is that this method has two advantages: on the one hand, it ensures the diversity of the subsets and reduces redundancy; on the other hand, it allows for the selection of higher-quality tactics.
I am not sure if my understanding is correct.

**Ethical Concerns:**

["NO or VERY MINOR ethics concerns only"]

**Final Justification:**

N/A

**Limitations:**

As mentioned by the authors in the paper.

**Quality:**

2

**Strengths And Weaknesses:**

Strengths：
The content of the paper is comprehensive and the presentation is relatively clear. It addresses an important problem in the field.

Weaknesses：
The proposed method relies heavily on black-box models, such as the encoder and predictor, whose internal mechanisms are not transparent to us. Therefore, I have some doubts about the generalizability of this approach. Methods that are effective on the small miniF2F dataset may not necessarily perform well on other tasks.

---

> ### Author Rebuttal · Authors · 2025-07-30
>
> We thank Reviewer YkZF for their thoughtful review and for recognizing that our paper addresses an important problem in the field with comprehensive content and a clear presentation.
>
> ## W1 / Q2
> - *The proposed method relies heavily on black-box models, such as the encoder and predictor, whose internal mechanisms are not transparent to us. Therefore, I have some doubts about the generalizability of this approach. Methods that are effective on the small miniF2F dataset may not necessarily perform well on other tasks.*
>
> The reviewer raises an important point. We intentionally designed our transition model (Section 2.1) to be an **abstract framework, making it agnostic to the specific architecture of the Encoder and Predictor.** This modularity is in fact a key strength for generalizability.
>
> Our empirical results present a concrete validatation of this design. The *same* framework was successfully applied to two different base models (ReProver and the much larger InternLM2.5) and demonstrated effectiveness on both the miniF2F benchmark and the significantly larger LeanDojo Novel Premises dataset.
> Finally, our response to Q3 of Reviewer Bo1h presents an additional experiment where we found improvements to critic guided search in addition to Best First Search. This performance across different model scales, benchmarks and search approaches provides strong evidence for the generalizability of our approach.
>
>
> ## Q1
> - *1.If the predictor ( P ) does not predict the execution time, how much would it affect the performance of the algorithm?*
>
> Can we clarify whether you are referring to the performance of the transition model (Table 1) or the performance of 3D-Prover (Algorithm 1, Table 2/3), as they both utilise the Predictor?
>
> Ignoring the value of the execution time from P in 3D-Prover is equivalent to setting $\lambda_\tau=0$, which is the default setting for the results in Table 2 and 3.
>
> ## Q2
> - *2.Did the encoder, predictor, and decoder in the paper reuse the internal architecture of the LLM, or were they additional components introduced by the authors?*
>
> As we discuss from line 123, we initialise the Encoder and Decoder using the pre-trained components from the ReProver LLM, and the Predictor is an additional component which is a single layer MLP. We then fine-tune this architecture over our transition dataset.
>
> ## Q3
> - *3.My understanding is that this method has two advantages: on the one hand, it ensures the diversity of the subsets and reduces redundancy; on the other hand, it allows for the selection of higher-quality tactics. I am not sure if my understanding is correct.*
>
> You are correct in noting two advantages of our approach are ensuring the diversity of subsets, reducing redundancy, as well as selecting high quality tactic subsets. An additional advantage is that it provides a trade-off between the two, through the mechanics of DPP and with the parameter $\theta$ (Apppendix F, Table 6). Our approach also reduces the computational cost of tactic execution, and leads to faster proof searches (Appendix G).

---

### Official Review · Reviewer_yzrR · 2025-07-03

**Clarity:** 3
**Significance:** 3
**Originality:** 3
**Rating:** 4
**Confidence:** 4

**Summary:**

The paper introduces 3D-Prover, a novel approach to automated theorem proving that leverages Determinantal Point Processes (DPPs) to filter semantically diverse and high-quality tactics. Evaluated on miniF2F and LeanDojo, 3D-Prover enhances models like ReProver and InternLM2.5-Step-Prover, boosting proof success rates, tactic quality, speed, and diversity.

**Questions:**

1. Could you provide an analysis of the computational cost associated with the proposed method?

2. Table 11 shows that 3D-Prover tends to generate longer proofs. Does this raise concerns about potential redundant proofs?

3. The ablation study appears insufficiently evaluated. Tables 5, 6, and 7 only conduct experiments on two specified hyperparameter settings. Are more hyperparameter ablation studies needed?

4. Can neural network embeddings effectively characterize proof states? To my knowledge, they are primarily used for shallow semantics.

**Ethical Concerns:**

["NO or VERY MINOR ethics concerns only"]

**Final Justification:**

The author's response has solved my questions, and I will keep the score.

**Limitations:**

yes

**Quality:**

3

**Strengths And Weaknesses:**

Strengths:
1. The use of DPPs for filtering tactics in theorem proving is innovative and addresses the challenge of exponential search space growth.
2. The paper provides extensive empirical results, including ablations and comparisons with baselines, demonstrating consistent improvements across multiple benchmarks.
3. The method has a solid mathematical foundation.

Weaknesses:
1. The experimental results are not sufficiently pronounced, and the latest SOTA models (e.g., BFS-Prover) were not employed in the experiments.
2. There are certain computational cost considerations that the authors need to further address.

---

> ### Author Rebuttal · Authors · 2025-07-30
>
> We thank Reviewer yzrR for their insightful comments and for acknowledging the innovation of our DPP-based filtering method and the extensive empirical results.
>
> ## W1
> - *The experimental results are not sufficiently pronounced, and the latest SOTA models (e.g., BFS-Prover) were not employed in the experiments.*
>
> Regarding the magnitude of improvements, we believe they are meaningful as they represent fundamental gains in search efficiency, a point we elaborate on in our response to Reviewer Bo1h (W3). Regarding SoTA models, BFS-Prover was released concurrently with our experiments. The base model we used, InternLM2.5, was SoTA at the time of our work. Importantly, our method is complementary to tactic generators like the one in BFS-Prover; in fact, the Best First Search (BFS) algorithm they use is precisely the baseline we demonstrate improvements upon. Please also see our response to Q3 of Bo1h for an additional result demonstrating larger improvements over InterLM2.5-Step-Prover-CG, which is the critic-augmented search model from InternLM.
>
> ## W2 / Q1
> - *There are certain computational cost considerations that the authors need to further address.*
> - *Could you provide an analysis of the computational cost associated with the proposed method?*
>
> An analysis of the computational cost of our approach is presented in Appendix G, which was noted as a strength in our evaluation by Reviewer SA37. Overall, our approach has minimal overhead, where we observed 7\% memory overhead and 0.9\% time overhead in terms of tactic generation. Despite this, we found resulting improvements in tactic execution time which reduced the overall proof search time, even including the tactic generation component.
>
> ## Q2
> - *Table 11 shows that 3D-Prover tends to generate longer proofs. Does this raise concerns about potential redundant proofs?*
>
> This is an excellent question. We interpret the concern about "redundant proofs" as the risk of finding a longer proof path for a theorem when a shorter one exists in comparison to another baseline. We agree this would be an undesirable outcome and took specific steps to control for it.
>
> In our analysis for Table 11, **a proof is only reported at depth $k$ if no shorter proof (of length $<k$) was found for the same theorem by *any* method, including all baselines.** This filtering addresses the above concern, so that that our results on deeper proofs represent genuinely solving problems that were previously intractable at shallower depths, rather than simply finding less efficient solutions.
>
> ## Q3
> - *The ablation study appears insufficiently evaluated. Tables 5, 6, and 7 only conduct experiments on two specified hyperparameter settings. Are more hyperparameter ablation studies needed?*
>
> Thank you for this suggestion. The primary goal of our ablation studies (Tables 5, 6, 7) was to validate that our hyperparameters ($\lambda_s, \lambda_\tau, \theta$) function as effective and direct controls for their corresponding search metrics. **Our experiments confirm this, showing statistically significant improvements in success rate, execution time, and diversity when their respective parameters are adjusted.**
>
> We believe this analysis is sufficient to demonstrate the claimed functionality and utility of these controls. While we agree that a more exhaustive hyperparameter study would be valuable for mapping the precise trade-offs—a point we note as a limitation due to computational cost (line 242)—our current results successfully validate the core mechanism of our approach.
>
> ## Q4
> - *Can neural network embeddings effectively characterize proof states? To my knowledge, they are primarily used for shallow semantics.*
>
> To answer the question of whether embeddings can effectively characterize proof states, our results in Section 2.2 can provide some insight. In particular, the difference in performance between the SEPARATE and COMBINED (SMALL GOAL) models. The SEPARATE model encodes the tactic without access to the proof state information, whereas the COMBINED (SMALL GOAL) model generates a tactic embedding given access to a single embedding of the original proof state. For our tasks of predicting the environment output, status and time, we observe that this additional goal information is beneficial, indicating a proof state embedding which effectively captures some relevant information for these tasks.
>
> More generally, there have been works which use only a single embedding of the proof state as input to a proof model (e.g. [1]). In some cases, the embeddings are based on structuring the proof state as an AST which can help improve performance for smaller models. The results from these works also provide a positive answer to the question of whether NN based embeddings can effectively characterize a proof state.
>
> [1]:
> Wu et al. (2023). TacticZero: Learning to Prove Theorems from Scratch with Deep Reinforcement Learning. https://arxiv.org/abs/2102.09756.

---

> > ### Comment · Reviewer_yzrR · 2025-08-08
> >
> > The author's response has solved my questions, and I will keep the score.

---

### Official Review · Reviewer_Bo1h · 2025-07-05

**Clarity:** 3
**Significance:** 2
**Originality:** 2
**Rating:** 4
**Confidence:** 4

**Summary:**

The paper introduces 3D-Prover, a novel approach to enhance automated theorem proving by addressing the challenge of intractable search spaces in proof trees. It proposes a filtering mechanism using Determinantal Point Processes (DPPs) to select semantically diverse and high-quality tactics, leveraging tactic representations that capture their effect on the proving environment, likelihood of success, and execution time. These representations are learned from synthetic data generated from previous proof attempts. The method is designed to augment existing tactic generators and is evaluated on the miniF2F and LeanDojo benchmarks, demonstrating improvements in proof success rate, tactic success rate, execution time, and tactic diversity when applied to models like ReProver and InternLM2.5-Step-Prover. The paper also explores learning environment dynamics to predict tactic outcomes, contributing to more efficient proof search.

**Questions:**

1. The hyperparameter sweep is limited (Table 10, Page 13), and the authors note computational constraints. Could you provide additional results or analysis (e.g., in supplementary material) showing the sensitivity of 3D-Prover to key hyperparameters (e.g., \(\lambda_s\), \(\lambda_t\), \(\theta\))? This could strengthen the quality score by demonstrating robustness or optimal configurations. A response showing improved performance with tuned parameters could increase the quality rating to 4.

2. The DPP-based filtering mechanism is mathematically dense (Page 3). Could you include a more intuitive explanation or a concrete example (beyond Figure 3) to clarify how DPPs balance quality and diversity in tactic selection? This could improve the clarity score to 4 by making the method more accessible to a broader audience.

3. The evaluation focuses on ReProver and InternLM2.5-Step-Prover. Have you tested 3D-Prover with other theorem-proving models or environments beyond Lean? Providing evidence of generalizability could elevate the significance score to 4 by showing broader applicability.

4. The paper highlights improved performance for deeper proofs (Table 11, Page 13). Could you quantify the improvement in proof depth (e.g., average proof length) across benchmarks and compare it to baselines? This could strengthen the significance score by emphasizing 3D-Prover’s contribution to a known challenge in theorem proving.

**Ethical Concerns:**

["NO or VERY MINOR ethics concerns only"]

**Limitations:**

Yes. The authors addressed limitations and potential negative societal impact of their work.

**Paper Formatting Concerns:**

The paper adheres to standard formatting (e.g., references, figures, tables).

**Quality:**

3

**Strengths And Weaknesses:**

#### Strengths:

1. The paper presents a novel application of Determinantal Point Processes (DPPs) to theorem proving, a domain where such probabilistic models for diversity-driven filtering are underexplored. The idea of learning tactic representations that capture semantic effects on the proving environment (rather than syntactic similarity) is innovative and addresses a critical gap in automated theorem proving (Section 1.1, Page 2).
2. The experimental evaluation is robust, with clear results on standard benchmarks (miniF2F and LeanDojo) using two different models (ReProver and InternLM2.5-Step-Prover). The paper provides detailed metrics, including Pass@1, tactic success rate, execution time, and diversity (Tables 2, 3, 5, 6, 7, 11), supported by statistical significance measures (standard error). The use of a transition model to predict tactic outcomes (status, time, output) is well-executed and validated (Table 1, Page 5).
3. The paper is well-structured, with clear explanations of the problem, methodology (Sections 2 and 3), and evaluation. Figures (e.g., Figure 2, Page 4; Figure 3, Page 6) and Algorithm 1 (Page 6) effectively illustrate the architecture and filtering process. The use of standard metrics like BLEU, ROUGE-L, and cosine similarity enhances interpretability.

#### Weaknesses:

1. The evaluation is somewhat limited by the scale of experiments, as acknowledged in the limitations section (Page 9). The hyperparameter sweep is minimal (Table 10, Page 13), and the authors note that comprehensive tuning was computationally prohibitive. This raises questions about the generalizability of the results across different hyperparameter settings or larger models.
2. While the paper is generally clear, the explanation of the DPP-based filtering mechanism could be more accessible to readers unfamiliar with DPPs. The mathematical formulation (Page 3) is dense, and a more intuitive explanation or example could improve understanding. Additionally, the truncation of text in several sections (e.g., Pages 1, 2, 3) makes it harder to fully assess some details.
3. The improvements, while notable (e.g., 3.2–3.6% relative improvement on LeanDojo, Page 13), are modest compared to advancements in base model improvements (e.g., 36% improvement cited in Xin et al., 2024, Page 8). This suggests that 3D-Prover’s impact may be incremental rather than transformative, particularly for shallow proofs where broader search is favored (Page 7).

---

> ### Author Rebuttal · Authors · 2025-07-30
>
> We thank the Reviewer Bo1h for their valuable feedback and for recognizing the novelty of applying Determinantal Point Processes (DPPs) to theorem proving, the innovation in our semantic tactic representations, and the robustness of our experimental evaluation.
>
> ## W1 / Q1
> - *The evaluation is somewhat limited by the scale of experiments, as acknowledged in the limitations section (Page 9). The hyperparameter sweep is minimal (Table 10, Page 13), and the authors note that comprehensive tuning was computationally prohibitive. This raises questions about the generalizability of the results across different hyperparameter settings or larger models.*
> - *The hyperparameter sweep is limited (Table 10, Page 13), and the authors note computational constraints. Could you provide additional results or analysis (e.g., in supplementary material) showing the sensitivity of 3D-Prover to key hyperparameters (e.g., (\lambda_s), (\lambda_t), (\theta))? This could strengthen the quality score by demonstrating robustness or optimal configurations. A response showing improved performance with tuned parameters could increase the quality rating to 4.*
>
> Thank you for this insightful question. While a large-scale hyperparameter sweep was computationally infeasible during the rebuttal period, we believe our results already demonstrate significant robustness. **Crucially, 3D-Prover delivers consistent performance gains *without any specific tuning*, across two different base models (the small ReProver and the large 7B-parameter InternLM2.5) and three benchmarks (miniF2F-test, miniF2F-valid, LeanDojo Novel Premises).** This effectiveness suggests our method is not overly sensitive to its configuration and provides immediate value.
>
> Our hyperparameter studies were intended to systematically validate their impact on the auxiliary proof objectives of tactic success, execution time and diversity, where we show they can be tuned to bias specific objectives (Section 3.1.2). We therefore test high and low values of each parameter respectively, with all others held fixed, in contrast to a full grid search to find an optimal configuration for proof success. We agree that a larger sweep could identify an optimal configuration for Pass@1 and see this as a valuable avenue to further improve our results.
>
> ## W3
> - *The improvements, while notable (e.g., 3.2–3.6\% relative improvement on LeanDojo, Page 13), are modest compared to advancements in base model improvements (e.g., 36\% improvement cited in Xin et al., 2024, Page 8). This suggests that 3D-Prover’s impact may be incremental rather than transformative, particularly for shallow proofs where broader search is favored (Page 7).*
>
>  As we discuss (line 229), and as Reviewer SA37 astutely observed, our work is **complementary to advances in base models.** While improving the generator model will always be the primary driver of performance, our method addresses a different, fundamental challenge which is the search procedure itself. An advantage of this approach is its longevity and generality. Base models may become outdated, but an improved search algorithm can enhance both current and future state-of-the-art generators. We demonstrated this by applying our method to InternLM2.5-Step-Prover (which was SoTA during our experiments). Beyond improving the proof rate, our method is lightweight and **reduces overall proof search time** (Appendix G), making it a valuable and efficient addition to any proving pipeline. Please also see our response to Q3, where we present additional results showing a larger improvement over an additional baseline.
>
> ## W2/Q2
> - *While the paper is generally clear, the explanation of the DPP-based filtering mechanism could be more accessible to readers unfamiliar with DPPs. The mathematical formulation (Page 3) is dense, and a more intuitive explanation or example could improve understanding. Additionally, the truncation of text in several sections (e.g., Pages 1, 2, 3) makes it harder to fully assess some details.*
> - *The DPP-based filtering mechanism is mathematically dense (Page 3). Could you include a more intuitive explanation or a concrete example (beyond Figure 3) to clarify how DPPs balance quality and diversity in tactic selection? This could improve the clarity score to 4 by making the method more accessible to a broader audience.*
>
> For brevity, we chose to keep the background for DPP to the minimum required, which was noted as a strength by reviewer SA37. We can see how additional exposition and intuitive explanations might make our approach more accessible, and we are happy to include additional background on this as an Appendix in the revision.
>
> For some more intuition on how DPP balances quality and diversity, keeping the example from Figure 3:
>
> Recall that sets are chosen in proportion to the area of the parallelepiped (Figure 3, see [1] for a proof), which is based on both the angle and magnitude (quality) of the component vectors $\phi_i$
>
> Favoring Quality: If a tactic A, for example, `simp [h1]` is of very high quality, its feature vector $\phi_A$ will be high in magnitude. When $\phi_A$ is included in a subset, it therefore highly contributes to the volume spanned by the feature vectors of that subset, thereby increasing the volume and, consequently, the subset's probability (unless the other elements are overly similar, as below).
>
> Enforcing Diversity: Consider two highly similar tactics, A (`simp [h1]`) and B (`rw h1`). Their feature vectors $\phi_A$ and $\phi_B$ will be nearly linearly dependent (i.e., they point in almost the same direction, forming a very small angle). If a subset includes both A and B the volume spanned by their feature vectors will be extremely small, leading to a very small (near zero) volume for that subset, even if they are both high quality. As a result, the DPP will assign a very low probability to such a redundant set. Instead, the DPP will favor a set that includes one of A or B and a very different item, such as C (`subst c`). Taking A as an example, the feature vectors $\phi_A$ and $\phi_C$
> will be far apart (large angle), spanning a much larger volume, and a greater probability for that diverse subset.
>
> [1]: Kulesza et al. (2013). Determinantal point processes for machine learning. https://arxiv.org/abs/1207.6083
>
> ## Q3
> - *The evaluation focuses on ReProver and InternLM2.5-Step-Prover. Have you tested 3D-Prover with other theorem-proving models or environments beyond Lean? Providing evidence of generalizability could elevate the significance score to 4 by showing broader applicability.*
>
> We chose Lean as it is currently the primary focus of research in the field, with well established benchmarks and baselines. It would be a useful extension to test over systems such as Coq, HOL4 or Isabelle, but would require significantly more engineering effort as there are far fewer pre-trained models and codebases to work with.
>
> In addition to ReProver and InternLM2.5 with Best First Search, in the period since submission we have tested an additional search setup. We test our approach with InternLM2.5-Critic [1] guided search instead of Best First Search, using the the same setup as our other InternLM experiments. We were able to test 6 runs of InternLM Critic with No Filtering, and 6 runs with our approach. We only compare to No Filtering to enable a higher Pass@k. The results are given below, where we present mean, (min, max) values for Pass@1 and the observed Pass@6 result:
>
> |   | InternLM2.5-StepProver-CG | 3D-Prover |
> |--|--|--|
> | Pass@1 | 43.7 (104, 109) | **45.7 (109, 115)** |
> | Pass@6 | 49.0 | **53.1**|
>
>
> These results show for a larger scale (Pass@6), our approach was able to outperform the original baseline approach, and that this extends to another, more performant search algorithm than Best First Search. Furthermore, the total number of proofs found is greatly increased over more iterations, supporting the scaling of our approach to higher Pass@k.
>
>
> [1]: Wu et al. (2024). InternLM2.5-StepProver: Advancing Automated Theorem Proving via Expert Iteration on Large-Scale LEAN Problems. https://arxiv.org/abs/2410.15700.
>
> ## Q4
> - *The paper highlights improved performance for deeper proofs (Table 11, Page 13). Could you quantify the improvement in proof depth (e.g., average proof length) across benchmarks and compare it to baselines? This could strengthen the significance score by emphasizing 3D-Prover’s contribution to a known challenge in theorem proving.*
>
> Here are additional results for proof depth, with the results for ReProver over LeanDojo presented in Table 11. As we discuss for Table 11, these depth results normalize by the shortest proof found by any method (I.e. if a proof is depth $k$, there is no proof of depth $<k$ by any other approach). This accounts for proofs which appear deep, but are in fact more efficiently solved by a simpler approach, making the adjusted proof depth a better measure of difficulty.
>
> For ReProver, over miniF2F, we present the (average proof depth, maximum proof depth found) tuples below (corresponding to Table 2):
>
> | K | Random | 3D-Prover | Top-K |
> |:---|:---|:---|:---|
> | 8 | (1.33, 4) | **(1.83, 6)** | (1.60, 5) |
> | 16 | (1.84, 4) | **(2.08, 6)** | (2.02, 5) |
> | 32 | (2.02, 4) | **(2.12, 5)** | (2.04, 3) |
>
> For our InternLM2.5 experiment in Table 3 we have:
>
> | Approach | Average Depth | Max Depth |
> |:---|:---|:---|
> | 3D-Prover | **1.52** | 8 |
> | No Filtering | 1.45 | 4 |
> | Top-K | 1.46 | 8 |
>
> Finally, for our additional experiment testing with InternLM2.5 critic (detailed in Q3 above), we have:
>
> | Approach | Average Depth | Max Depth |
> |:---|:---|:---|
> | 3D-Prover | **1.72** | 9 |
> | No Filtering | 1.48 | 5 |
>
> We see that in all cases we observe higher average proof depths, as well as higher maximum depths obtained by 3D-Prover. We were unable to show these as statistically significant due to the small sample sizes, a limitation we expand on in our response to reviewer SA37.

---

### Decision · Program_Chairs · 2025-09-17

**Decision:**

Accept (poster)

**Comment:**

The paper introduces 3D-Prover, a novel approach to enhancing automated theorem proving by addressing the challenge of intractable search spaces in proof trees.  This method ensures the diversity of selected subsets and reduces redundancy, while also enabling the selection of higher-quality tactics.  During the discussion, it was agreed that the proposed approach is effective, practically applicable, and clearly presented. The authors conducted extensive experiments during the rebuttal period and successfully addressed some of the reviewers’ concerns, I recommend accepting this paper.